

# Geostrophic circulation and tidal effects in the Gulf of Gabès

Maher Bouzaiene[1], Antonio Guarnieri[1], Damiano Delrosso[1], Ahmad F. Dilmahamod[2], Simona Simoncelli[1], Claudia Fratianni[1]

[1]Istituto Nazionale di Geofisica e Vulcanologia, Sezione di Bologna, Bologna, Viale Berti Pichat 6/2, 40127, Italy
[2]GEOMAR Helmholtz Centre for Ocean Research Kiel, Wischhofstr 1-3, 24148 Kiel, Germany

*Correspondence to*: Maher Bouzaiene (maher.bouzaiene@ingv.it)

**Abstract.** The mean kinematic features in the Gulf of Gabès region is analyzed based on 30 years of altimetry data (1993-2022) and the outputs of a high resolution ocean model for the year 2022. A comparison of the seasonal variability in three different geographical areas within the gulf is presented. In the northern and southern parts of the gulf, anticyclonic

structures prevail, while the central area is dominated by divergence. Similarity in the flow topology is found in these three areas of the gulf due to the signature of hyperbolic regions. In winter and fall, the mean flow is oriented northward while it is reversed in spring and summer. The tidal perturbation influences sea level, kinetic energy and hyperbolic geostrophic structures, leading to the generation of a cyclonic current located in the central part of the gulf and to the presence of persistent strain gradients amplifying hyperbolic structures. The Finite Time Lyapunov Exponent (FTLE) computed using

altimetry data highlights the link between physical and biogeochemical processes, with the Gulf of Gabès mean circulation features acting as transport barriers for phytoplankton dispersion.

## 2   Introduction

Tidal forcing plays a crucial role in ocean circulation, and the Gulf of Gabès (GG) is notably influenced by this phenomenon. Understanding tidal interactions with complex dynamics and their impact on the transport of passive and active tracers (such

as pollutants and marine species) is challenging in this area. Well-known as a region of relevant tides with a semi-arid climate, high temperature, relatively high salinity and strong density gradients, the GG is located in the southern part of the Sicily Channel and represents a site of water masses exchange between the western and the eastern Mediterranean Sea basins.

The gulf is considered one of the richest areas of the Mediterranean Sea in terms of nutrients availability and biological

production. Due to the gentle slope of its continental shelf and its shallow depth (Figure 1), the GG has the highest tidal range (with difference between high and low tide up to almost 2 m) in the Mediterranean Sea, which strongly influences its circulation (Abdennadher and Boukthir, 2006; Sammari et al., 2006; Poulain and Zambianchi, 2007). Apart from the tides, anticyclonic winds are also one of the drivers of the GG circulation (Sammari et al., 2006). Originated from the Atlantic Water (AW), the Atlantic Tunisian Current (ATC) and the Bifurcation Atlantic Tunisian Current (BATC) are the strongest



surface currents evolving in the gulf (Ben Ismail et al., 2015), while the Atlantic Ionian Stream (AIS) is generated by current instability and topography (Menna et al., 2019), and it flows North of Malta island without penetrating in the GG (Figure 2a) (Pinardi et al,. 2015; Bouzaiene et al., 2020). The ATC is a permanent surface current characterized by low salinity (Sammari et al., 1999). It crosses the Strait of Sicily and circulates along the Tunisian coast (Sorgente et al., 2011) where it splits into two branches. One of them interests the coastal strip and is directed southward (Ben Ismail et al., 2010). Due to its

lower salinity, ATC branch circulates further offshore from the coast in a south-eastern direction where it eventually follows the Libyan plateau (Figure 2a) (Millot and TaupierLetage, 2005). This branch, observed especially in winter, is called the Atlantic Libyan Current (ALC). It circulates along the Libyan shelf break where the mean flow is represented by a weak current bounded by a cyclonic vortex referenced as the Libyan Shelf Break Vortex (LSBV, Sorgente et al., 2011, see Figure 2a). The BATC continues to circulate offshore, generating the Medina Gyre (MG) and the Southern Medina Gyre (SMG)

(Figure 2a), whose formation is due to current instability or/and topography (Jouini et al., 2016, Menna et al., 2019).
The GG dynamics is characterized by small and large-scale inter-annual and seasonal variability in the surface layer (Jebri et al., 2016). This system is identified by many spatial-temporal structures interacting with each other and producing an extremely complex and variable circulation. Currents, filaments and eddies are responsible for water mass transport, thus the understanding of their seasonal and inter-annual variability is crucial for a wide variety of reasons, such as regional water

exchanges with the open sea, large scale turbulent flow spreading, propagation of particles and dispersion of sediments in the coastal zones. This is even more important in the GG environment where the anthropogenic pressure has dramatically increased due to coastal phosphate processing plants by-products, frequent oil spill episodes, general pollution factors such as floating marine debris or plastic, micro and macro litter (Ben Ismail et al., 2022). In addition, Lagrangian studies on the transport of nutrients, jellyfish, eggs, and larvae would benefit from a better knowledge of the mean circulation variability.

To provide a realistic study for the GG circulation it is necessary to evaluate in detail the long-term variability and the influence that tidal forcing, interacting with topography, exert on the system, resulting in the generation of new structures influencing turbulence and circulation in the entire gulf.
It has been shown in Elhmaidi et al. (1993) that turbulent features induce discrepancy between modeled and theoretical dispersion laws in case of two-dimensional turbulent dispersion theory for isotropic and homogeneous flow. Two anomalous

absolute dispersion (5/3, elliptic) and (5/4, hyperbolic) power laws were found in previous studies (Bouzaiene et al., 2021). These anomalous regimes have been related to the sea topology through the presence of elliptic and hyperbolic structures (Bouzaiene et al., 2018, 2021). In the eddy inner parts, these areas are referred as elliptic regions characterized by high vorticity gradients, while hyperbolic features are detected in the coherent structure outer parts and can be related to shared/stretched ocean flow.

To our knowledge, the impact of tides on topology (the distribution of elliptic and hyperbolic regions) in the GG has not been studied before, even though tidal forcing is very important in this area. Hence, given their potential influence on phytoplankton blooms, nutrient distribution, and marine litter dispersion, it is essential to gain a better understanding of how tides influence the circulation, dynamics, and sea topology.





Altimetry data analysis allows analyzing the geostrophic circulation and the kinematic properties of mesoscale structures.
However, their low temporal and spatial resolutions do not allow performing a realistic study on the dynamics introduced by tides. In order to address this issue, we leveraged high temporal resolution (hourly) model outputs from a numerical system which includes tides.

In this study, we focus on the kinematic properties of the geostrophic component of the circulation in the GG and on how tides affect currents. We do this by using altimetry data covering the time period 1993-2022 and model analysis data for the
year 2022, both distributed by the Copernicus Marine Service (CMS, http://marine.copernicus.eu/). Previously, some efforts to understand this oceanographic system have been made by focusing on the general aspects of the Gulf of Gabès circulation. The comprehension of how tides influence the geostrophic features is still an open question.

Our aim is to investigate new features of the geostrophic circulation, estimated as the balance of the Coriolis force and the horizontal pressure gradients, and to assess the impact of tides on the geostrophic circulation. High resolution ocean
circulation modeling and satellite altimetry could enhance our understanding of the geostrophic transport. The paper is organized as follows: in section 2 we describe the datasets used and the methods applied. The results on geostrophic structures from altimetry and model data are presented in section 3. Summary and conclusions are proposed in section 4.

## 3    Material and methods

### 3.1 Datasets
The geostrophic circulation in the GG has been investigated by means of remote-sensed altimetry data and outputs from a high resolution oceanographic numerical system.

### 3.1.1    Altimetry data
The satellite altimetry dataset used in this study is a subset of the CMS SEALEVEL_EUR_PHY_L4_MY_008_068 product (European Union-Copernicus Marine Service. (2021). EUROPEAN SEAS GRIDDED L4 SEA SURFACE HEIGHTS AND
DERIVED    VARIABLES    REPROCESSED    (1993-ONGOING)    [dataset].    Mercator    Ocean    International. https://doi.org/10.48670/MOI-00141), covering a 30-year period (1993–2022) with a spatial resolution $0.125° \times 0.125°$. The variable used is the absolute surface geostrophic velocity, while altimetry data were used to estimate the vorticity, the divergence, the Okubo-Weiss parameter, the deformation gradients and the Finite Time Lyapunov Exponent (FTLE). These parameters were investigated in order to elucidate the mean circulation, persistent currents, eddies and gyres.

### 3.1.2    Chlorophyll a data
The chlorophyll a dataset used in this study is the CMS OCEANCOLOUR_MED_BGC_L4_MY_009_144 product. We used the daily mass concentration of chlorophyll a in sea water (CHL) at 1 km resolution from the Ocean Satellite Observations for multi-years Bio-Geo_Chemical (BGC) regional datasets (https://doi.org/10.48670/moi-00300.

### 3.1.3    Model data



Sea Surface Height (SSH) fields from the CMS MEDSEA_ANALYSISFORECAST_PHY_006_013 product (Clementi et al., 2021) covering the year 2022, are used to compute geostrophic currents. We have chosen year 2022 since at the time the dataset was processed it was the only complete year for the CMS system including tidal signal in the hydrodynamic model used. The physical component of the Mediterranean Sea within the framework of CMS (Med-Physics) is a tidal, coupled hydrodynamic-wave model with a data assimilation system implemented over the whole Mediterranean Sea, with a

horizontal resolution of 1/24° (~4 km) and 141 unevenly spaced vertical z* levels (Clementi et al. 2017). More detailed information on the system and its products can be found in the Quality Information Document (at https://catalogue.marine.copernicus.eu/documents/QUID/CMEMS-MED-QUID-006-013.pdf).

### 3.2 Methods

To describe the kinematic properties and the circulation of the GG we estimate the geostrophic currents for the year 2022

from the MEDSEA_ANALYSISFORECAST_PHY_006_013 product SSH fields and we then compute the four following quantities from the altimetry data and model data (described in section 2.1.1 and 2.1.3, respectively): normalized vorticity, normalized divergence, normalized Okubo-Weiss parameter and FTLE.

### 3.2.1     Estimation of the geostrophic currents

The model SSH field (section 2.1.3) was used to estimate the geostrophic currents, resulting from the balance between the

Coriolis force and the horizontal pressure gradient. The zonal (ugeos) and the meridional (vgeos) components of the geostrophic velocities are derived from the geostrophic equations as follows (Vigo et al., 2018a; 2018b):

$$\text{ugeos} = \frac{-g}{f}\frac{\partial \eta}{\partial y} \quad (1)$$

$$\text{vgeos} = \frac{g}{f}\frac{\partial \eta}{\partial x} \quad (2)$$

Where, x and y are the longitude and the latitude components respectively, $\eta$ is the sea surface elevation, g=9.81 m/s$^2$ is the gravity acceleration, f=2$\Omega$sin($\lambda$) is the Coriolis parameter, $\lambda$ is the latitude in degrees and $\Omega$=2$\pi$/T is the Earth angular velocity, being T the period of rotation.

In order to evaluate the tidal residual from the full SSH signal, a Doodson filter was applied to the dataset, following the approach proposed in the Manual on Sea Level Measurement and Interpretation of the IOC (1985). The Doodson is a low-pass, symmetric filter based on the definition of 19 coefficients as follows:

F(t)= (2, 1, 1, 2, 0, 1, 1, 0, 2, 0, 1, 1, 0, 1, 0, 0, 1, 0, 1); F(t)=F(-t)

The value of the de-tided sea level $SSH_{res}$ at time $t_0$ is calculated as:

$$SSH_{res}(t_0) = \frac{1}{30}\sum_{d=-19}^{d=19} F(d)H(t_0 + d); d \neq 0 \quad (3)$$





Where H denotes the sea level elevation, $t_0$ is the time expressed in hours and the coefficients d represent the increasing or decreasing hours with respect to the central value $t_0$.

### 3.2.2  The normalized vorticity

The normalized vorticity is defined as (Poulain et al., 2023):


$$\zeta^* = \frac{\zeta}{f}, \zeta = \frac{\partial v}{\partial x} - \frac{\partial u}{\partial y} \quad (4)$$

where $\zeta$ is the relative vorticity, which is a good indicator of sub-mesoscale, mesoscale, filaments, eddies and fronts activity in the ocean. If $\zeta^* \sim O(1)$ the circulation is driven by mesoscale processes, the flow shows a-geostrophic features and the relative vorticity is balanced with the planetary vorticity. If $\zeta^* > O(1)$ the flow is characterized by sub-mesoscale driven circulation.

### 3.2.3  The normalized divergence

The normalized divergence, a fundamental metric to characterize the transport of passive and active tracers, is defined as (Poulain et al., 2023):

$$\delta^* = \frac{\delta}{f}, \delta = \frac{\partial u}{\partial x} + \frac{\partial v}{\partial y} \quad (5)$$

where $\delta$ is the horizontal divergence of the velocity field. It allows to detect two different dynamical oceanic zones: for

$\delta^* > 0$ (divergence) the flow fields tend to propagate outward through the surrounding surface of a closed control volume, diverging from its center, whereas for $\delta^* < 0$ (convergence) the flow particles tend to converge to the center of the volume.

### 2.2.4 The normalized Okubo-Weiss parameter

The normalized Okubo-Weiss parameter serves as a powerful indicator to distinguish between two different topological

domains: elliptic or hyperbolic. It is defined as:

$$Q^* = \frac{\left(S^2 - \zeta^2\right)}{\left(S^2 + \zeta^2\right)} \quad (6)$$

Where S is the strain or rate of deformation of the flow and it is composed by a shear term $S_s$ and by a normal term $S_n$. It is defined as follow:

$$S = \left[S_s{}^2 + S_n{}^2\right]^{\frac{1}{2}} \quad (7)$$

Where $S_s$ and $S_n$ are defined as


$$S_s = \left(\frac{\partial v}{\partial x} + \frac{\partial u}{\partial y}\right)^2, S_n = \left(\frac{\partial u}{\partial x} - \frac{\partial v}{\partial y}\right)^2$$

Where $Q^* \sim -1$ an elliptic domain can be defined, while where $Q^* \sim 1$ a hyperbolic region can be identified (Okubo, 1970, Weiss, 1991, Elhmaidi et al., 1993; Bouzaiene et al., 2018, 2021). As shown by Bouzaiene et al. (2021), the flow dynamical properties in the eddy inner parts (elliptic regions) and surrounding coherent structures (hyperbolic structures) are very different. In this study S is normalized by f to identify the sheared and/or stretched regions:



$$S^* = \frac{S}{f} \quad (8)$$


### 3.2.4 The Finite Time Lyapunov Exponent

The Finite Time Lyapunov Exponent (FTLE) $\lambda_t$ is a parameter which describes the separation amongst particles in a specific time interval and it has been used in several ocean applications to identify the Lagrangian Coherent Structures (LCS) (Shadden et al., 2005; Farazmand and Haller, 2012). In previous investigations within the Mediterranean region, the

emphasis was on the Finite Scale Lyapunov Exponent (FSLE) rather than the FTLE. The FSLE was calculated on the basis of the exponential growth of distances between Lagrangian particle pairs initially separated. This calculation served the dual purpose of identifying Lagrangian Coherent Structures (LCS), as demonstrated by d'Ovidio et al. (2004, 2009), and comparing LCS with Lagrangian surface drifter trajectories, as explored by Bouzaiene et al. (2020).

More recently, Morales-Márquez et al. (2023) investigated the use of FSLE to characterize LCS concerning mixing and
transport properties in the upper layer of the entire Mediterranean Sea. Despite these advancements, the application of the parameter $\lambda_t$ in coastal Mediterranean zones, particularly in the GG, remains an unanswered question. This study seeks to address this gap by computing $\lambda_t$ specifically for LCS analysis in these areas.

FTLE is a local scalar that represents the separation rate of initially neighboring particles for a finite time $[t_0, t_0+T]$. At position $x_0$ and time $t_0$, $\lambda_t$ is defined as follows (Haller, 2002, 2015; Liu et al., 2018):

$$\lambda_t(x_0, t_0, T) = 0.5 \left[ \frac{\log \lambda_{max} \left( \left[ \frac{\partial \Phi(x_0, t_0+T, t_0)}{\partial x_0} \right]^{tr} \left[ \frac{\partial \Phi(x_0, t_0+T, t_0)}{\partial x_0} \right] \right)}{T} \right] \quad (9)$$

where $\lambda_{max}$ is the largest eigenvalue of the Cauchy-Green stress tensor, while the flow field of fluid particle trajectories is defined as $\Phi(x_0, t_0 + T, t_0)$ and indicates the matrix transpose.

In 2D turbulence theory, the eigenvalues of the Cauchy-Green tensor quantify the stretching of fluid particles along their relevant directions (Liu et al., 2018). In this work, we compute the positive-time (T>0) of $\lambda_t$ field. This eigenvector is called "forward Finite-Time Lyapunov Vector".

It has been shown that $\lambda_t$ is predominantly reliable to capture coherent structures starting from an integration time of 6 days, with no upper limit (Du Toit, 2010; Rypina et al., 2011; Liu et al., 2018). In this study, $\lambda_t$ is averaged for time integrations of 30 years to detect the mean features and over 7 days to compare it to phytoplankton blooms occurring in GG. Here, $\lambda_t$ is calculated from the velocity fields derived from satellite altimetry data at temporal and spatial resolutions of 1 day and 1/8°, respectively. Our choice is to set the resolution of the mean initial trajectory conditions to 800 meters ×800 meters,

corresponding to 1/128°, about 16 times larger than the velocity field resolution, which guarantees an LCS accurate enough for capturing oceanic features (Onu et al., 2015). In general, high values of $\lambda_t$ indicate the edges of coherent structures, fronts and filaments (hyperbolic regions), while low values correspond to the inner parts of the eddies (elliptic areas). Both are considered as transport barriers (Blazevski and Haller, 2014).





# 4    Results

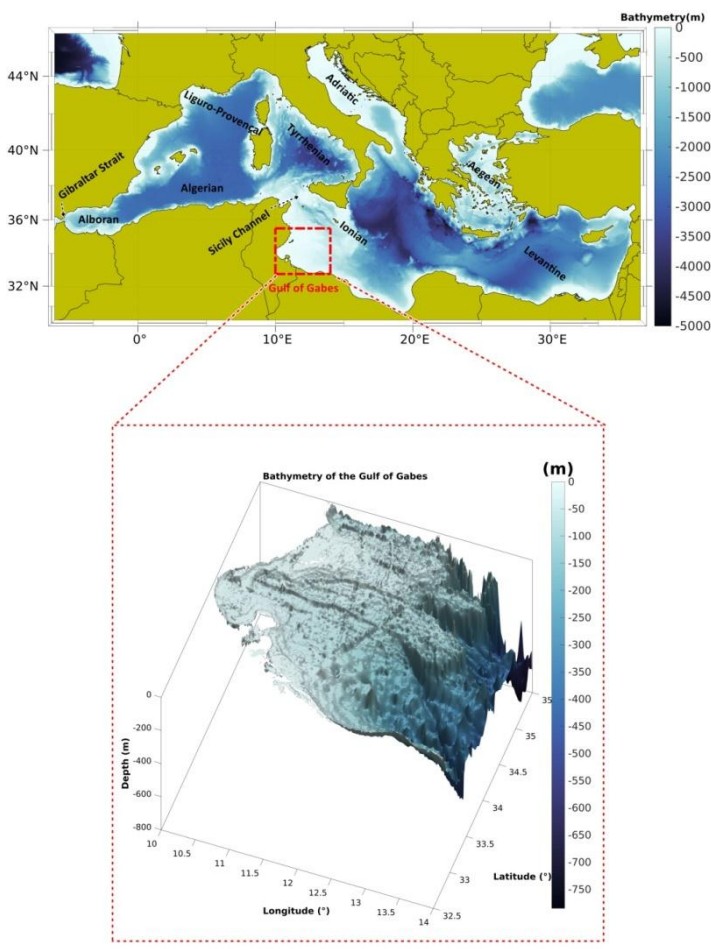

**Figure 1:** Bathymetry of the Mediterranean Sea with the main geographical sub-basins and straits locations. The red dashed rectangle in the upper panel shows the geographical limits of the larger Gulf of Gabès domain. A zoom of the bathymetry in the Gulf of Gabès is shown in the lower panel. The bathymetry was derived from the Global Earth Bathymetric Chart of the Oceans for the 2022 version with a spatial resolution of ~0.45 km (GEBCO_2022, https://www.gebco.net).

Three subareas were identified in the GG (black boxes in Figure 2) in order to highlight differences and similarities in the dynamical features: Northern Gulf of Gabès (NGG, $11^{o}$E-$12^{o}$E and $34.6^{o}$N-$35.25^{o}$N), Central Gulf of Gabès (CGG, $10^{o}$E-$11^{o}$E and $33.75^{o}$N-$34.4^{o}$N) and Southern Gulf of Gabès (SGG, $11.1^{o}$E-$12.2^{o}$E and $33^{o}$N-$33.75$N$^{o}$). 30 years of satellite altimetry data are used to overview the mean kinematic features in the GG domain and in the three specified subareas, as detailed in section 3.1.



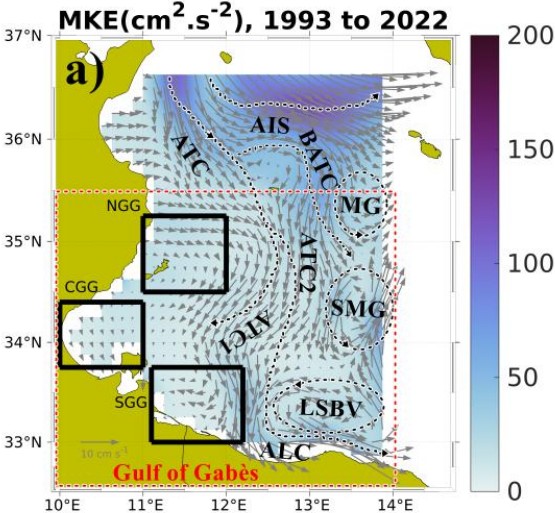

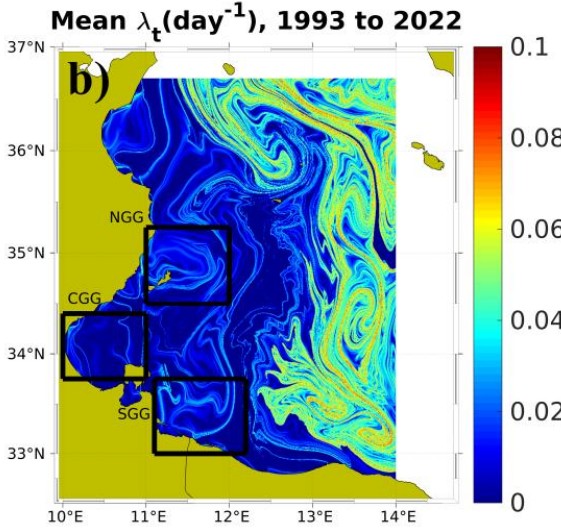


**Figure 2:** (a) Mean Kinetic Energy (MKE) with superimposed mean geostrophic currents estimated over the period 1993-2022; (b) Mean Finite Time Lyapunov Exponent ($\lambda_t$) computed from geostrophic velocities from altimetry data (SEALEVEL_EUR_PHY_L4_MY_008_068) in the same period. Three subareas are selected close to the coastal areas of the gulf as follow: NGG: Northern Gulf of Gabès, CGG: Central Gulf of Gabès, SGG: Southern Gulf of Gabès. List of acronyms of the main circulation features: MG: Medina Gyre, SMG: Southern Medina Gyre, LSBV, Libyan Shelf Break Vortex, ATC1, Fist Atlantic Tunisian Current, ATC2: Second Atlantic Tunisian Current, ALC: Atlantic Lybian Current, AIS: Atlantic Ionian Stream, BATC: Bifurcation Atlantic Tunisian Current.



The relatively low temporal and spatial resolutions of the data do not allow to make considerations on the impact that tides

have on GG features in a smaller area. Hence, we focused on the GG Smaller Domain (GGSD, 10.00°E-11.35°E and
33.40°N-34.75°N) where we derived geostrophic currents from the SSH model fields allowing us to test the tides impact in
the GG, as explained in section 3.2. SSH spatial and temporal averages in the GGSD have been removed from the native
SSH fields.

The resulting mean seasonal geostrophic circulation in the GGSD was compared to its counterpart computed from the de-

tided SSH fields, in order to assess the impact of tides on the geostrophic dynamics of the Gulf. The de-tiding on the native
SSH fields was performed applying a Doodson filter (see section 2.2.1 for details).

## 4.1 Altimetry data analysis of the mean geostrophic circulation

Figure 2a shows the geostrophic circulation (grey arrows) averaged over the period 1993-2022, superimposed to the Mean
Kinetic Energy MKE = $\langle 0.5(\text{ugeos}^2 + \text{vgeos}^2)\rangle$ for the same period, where <…> represents the average over the 30 years.

Several well-known structures are clearly visible, namely: (1) the edges of the cyclonic features referenced as the Medina
Gyre (MG), the Southern Medina Gyre (SMG) and the Libyan Shelf Break Vortex (LSBV); (2) the first Atlantic Tunisian
Current (ATC1) which is well developed along the Tunisian coasts; and (3) the second Atlantic Tunisian Current (ATC2),
flowing near the Libyan boundaries and forming the Atlantic Libyan Current (ALC), in agreement with the results of
Sorgente et al. (2011), Jebri et al. (2016) and Menna et al. (2019).

The MKE shows the presence of energetic features flowing into the GG as well as surrounding eddies, BATC, ATC1 and
ATC2 with MKE of ~50-100 cm$^2$/s$^2$, while the Atlantic Ionian Stream (AIS) inflows through the north-eastern Ionian Sea
with a maximum MKE of approximately 100-150 cm$^2$/s$^2$. The mean FTLE averaged over the 30-yearaltimetry data period
($\lambda_t$; Equation 9) is shown in Fig. 2b. $\lambda_t$ can be applied to investigate the link between the chaotic turbulence and the
chlorophyll concentration, which are known to be related by inverse proportionality. Intense stirring induced by strong

turbulence disperses the high input of nutrients when uplifted from deeper layers, whereas larger amounts of nutrients remain
in more quiescent zones (decreased turbulence; Hernandez-Garcia et al., 2010). The regions located far from the GG, are
characterized by a large $\lambda_t$, close to 0.1 day$^{-1}$, indicating strong chaotic advection clearly evidenced by the presence of
intense eddies and persistent currents (MG, SMG, LSBV, AIS, see Figure 2a vs. 2b). These features appear as local barriers
to transport, inhibiting the biological production. On the contrary, in the coastal zones of the GG, $\lambda_t$ tends to zero. These

zones may favor the nutrient standing stocks due to the weak effect of the horizontal mixing and stirring. This could be one
of the causes of the observed high chlorophyll concentration close to the Tunisian boundaries as shown in previous studies
(Bel Hassen et al., 2010; Macias et al., 2018; Kotta et al., 2019), where nutrients would flee from high turbulent zones to
settle in less chaotic areas.

The NGG is clearly evidenced by the presence of large anticyclonic current with $\lambda_t$~0 in its core, while $\lambda_t$ is greater than zero

in the eddy outer part. A similar pattern can also be observed in the SGG. On the contrary, the mean FTLE in the CGG
suggests the presence of filaments and fronts (Fig. 2b). Normalized vorticity ($\zeta^*$), normalized Okubo-Weiss ($Q^*$),
normalized divergence ($\delta^*$), and normalized deformation ($S^*$) over the 30 years (1993-2022) from the altimetry product are



shown in Figure 3a,b,c and d, respectively, confirming the presence of the MG, SMG and LSBV eddies. Except for the MG, within the interior of these structures, the value of Q* is negative (elliptic regions) due to high vorticity gradients, whereas in

the surrounding coherent structures the value of Q* is positive, with predominantly hyperbolic areas due to strong deformation gradients (Fig. 3b).

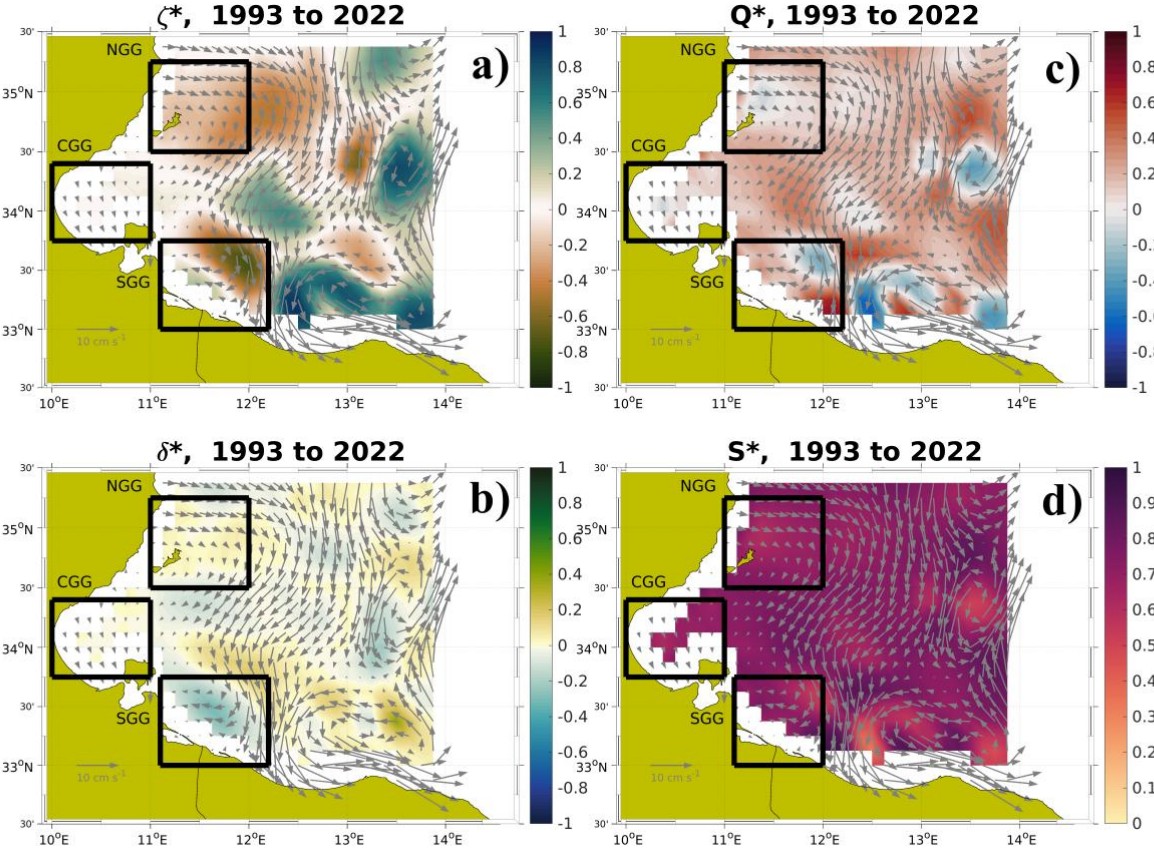

**Figure 3:** Mean circulation from altimetry data (SEALEVEL_EUR_PHY_L4_MY_008_068) estimated over the period 1993-2022 (a) Normalized vorticity (ζ*), (b) normalized Okubo-Weiss parameter Q*, (c) the normalized divergence (δ*) and (d) normalized deformation (S*) with superimposed mean geostrophic velocities. Three subareas are selected close to the coastal areas of the gulf as follow: NGG: Northern Gulf of Gabès, CGG: Central Gulf of Gabès, SGG: Southern Gulf of Gabès.

The vorticity mostly agrees with the divergence (Fig. 3a and c) for ζ* and δ* higher than zero, implying the presence of several cyclonic eddies where the flow tends to propagate outward through the surrounding eddy cores (divergence).

Upwelling of deep, nutrient-rich water masses occur in these areas, leading to enhanced biological production at the surface. In the opposite scenario (negative ζ* and δ*) the surface flow is pointing towards the inner parts of the anticyclonic eddy (convergence), pushing water towards its center of mass, then sinking to the bottom layers. The flow is sheared or stretched (S*~O(1)) in the eddy outer parts where the current is very unstable. These zones can be identified as hyperbolic regions due



to strong deformation gradients, while for S*~0 (inside coherent vortices) the rotation is dominant (Figure 3 b and d). In order to compare the dynamics of regions relatively close to the GG with farther ones, ζ*, δ* and Q* are seasonally evaluated in the three sub areas displayed in Figure 4. They can all be classified as hyperbolic regions (Q*>0, magenta lines in Figure 4) meaning that the flow can be stretched or sheared. Except for some seasons, the NGG and CGG surface waters tend to rotate into gyres (elliptic areas, Q*<0). In the CGG, the vorticity is oscillating from ~-0.5 (anticyclonic) to ~0.5 (cyclonic), while ζ* shows mostly negative values in the two other areas considered (thus indicating the presence of anticyclonic vortices). In agreement with the vorticity values, the divergence is negative in the SGG (convergent flow, Fig 4c). In the NGG and CGG, δ* indicates positive values, mostly greater than zero in the NGG, thus denoting the presence of upwelling flows (Fig. 4a,b blue line). The difference in the divergence of the three subareas might be related to the different interaction of the main forces (i.e tides and winds) with the bottom topography.

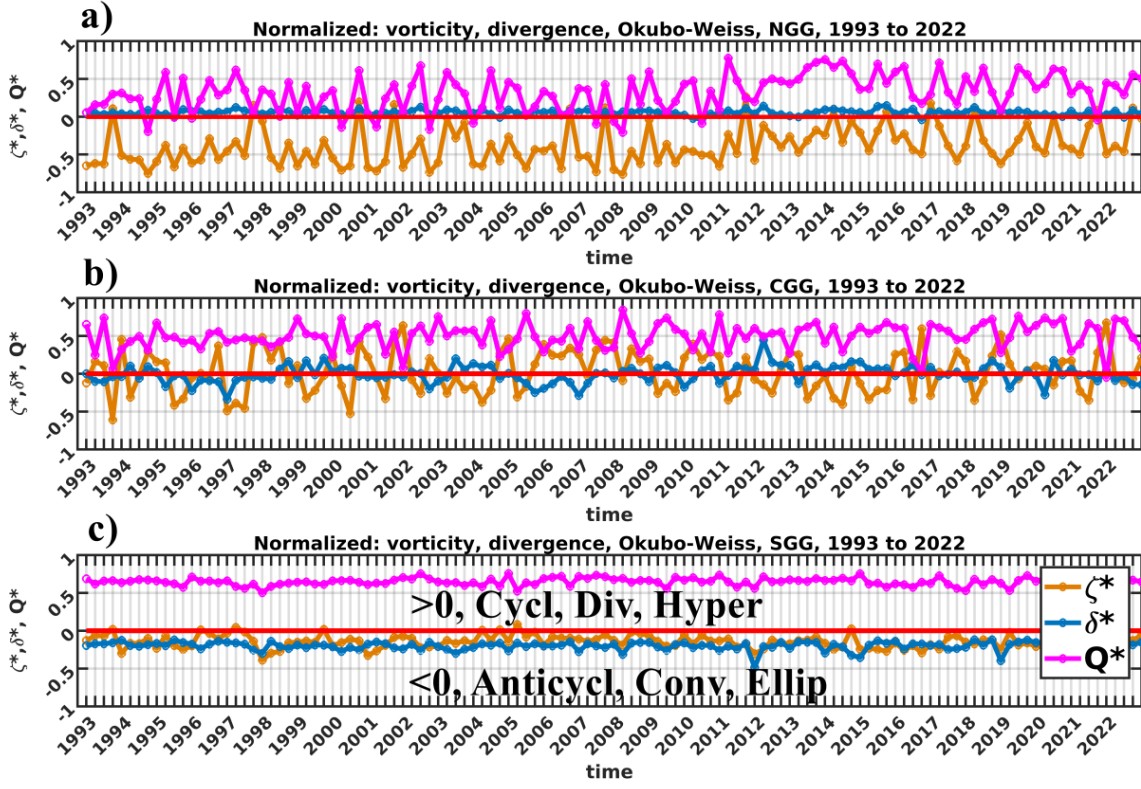

**Figure 4:** Time series of normalized vorticity (ζ*; yellow), normalized divergence (δ*; blue) and normalized Okubo-Weiss parameter (Q*, magenta) estimated from geostrophic velocities provided by altimetry data (SEALEVEL_EUR_PHY_L4_MY_008_068) in the Gulf of Gabès over the period 1993-2022. Panel (a): North GG; Panel (b): Central GG; Panel(c): South GG.

### 3.2 Model data analysis of the geostrophic circulation in 2022





The GG surface circulation is strongly controlled by tides (Zayen et al., 2020). The highest tidal ranges can be detected in the central part of the GG, whilst they are much less significant outside of the gulf (Abdennadher and Boukthir, 2006). Tidal movements, which induce vertical mixing, could be considered as a potential cause of high chlorophyll concentration in the CGG (Macias et al., 2018). The following sections will focus on the GGSD dynamics, both because of its importance from the biological production point of view, as shown previously by Feki-Sahnoun et al. (2018), and because it represents a site

of particular interest to investigate the influence of tidal forcing on dynamics, given that GGSD is one of the areas in the Mediterranean Sea where the highest tides can be found (Othmani et al., 2017).

### 3.2.1    Seasonal variability

The comparison of the mean seasonal geostrophic circulation in 2022 derived from full SSH and detided SSH is shown in Figure 5, where the arrows representing the geostrophic velocities are superimposed to the kinetic energy. During winter/fall

the mean flow tends to inflow from the south to the north, while in spring and summer its circulation is mainly cyclonic bordering the coastline. Our results are in good agreement with previous studies in the Mediterranean Sea (Vigo et al., 2018a). The difference in flow direction in the GGSD can be related to the topography of the gulf (Figure 1) and/or to the horizontal pressure force influenced by anomalous cyclonic and anticyclonic atmospheric conditions which were present in 2022, as found in Marullo et al. (2023). These atmospheric conditions could also be potential causes of the different seasonal

geostrophic patterns shown in Figure 5. The influence of tides can also be found in the difference of the seasonally averaged KE computed in the case of full SSH and detided SSH. In winter, in the case of geostrophic currents computed from full SSH, the KE can reach ~50 $cm^2/s^2$, while it decreases in the other seasons with the lowest values detected in summer/fall (KE $<5$ $cm^2/s^2$). The tides influence on geostrophic circulation can be also quantified by the differences in mean KE computed from full SSH and detided SSH fields. The largest MKE values (8.25 and 8.06 $cm^2/s^2$ for tidal and detided fields,

respectively) can be observed in winter, while the lowest values are observable in fall, with an average value of about 0.17/0.14 $cm^2/s^2$. In spring and summer, the MKE values of 2.06/1.3 and 0.43/0.23 $cm^2/s^2$, respectively. Moreover, in spring and summer the relative weight of the tidal component of the KE is much stronger (46% and 37% respectively) than in winter and fall (2% and 17%). Since tidal forcing itself does not have significant seasonal variability, it is clear that it does not affect the varying seasonal patterns shown in Figures 5a-d. The impact of tides on geostrophic circulation can be

observed for the cyclonic vortex detectable in spring, summer and fall north-west of Djerba Island (~$10.4E^o$-$10.9E^o$ and ~$33.8N^o$-$34.15N^o$, see the black rectangle in Fig 5) in the full SSH field, which disappears when deriving geostrophic circulation from detided SSH fields.





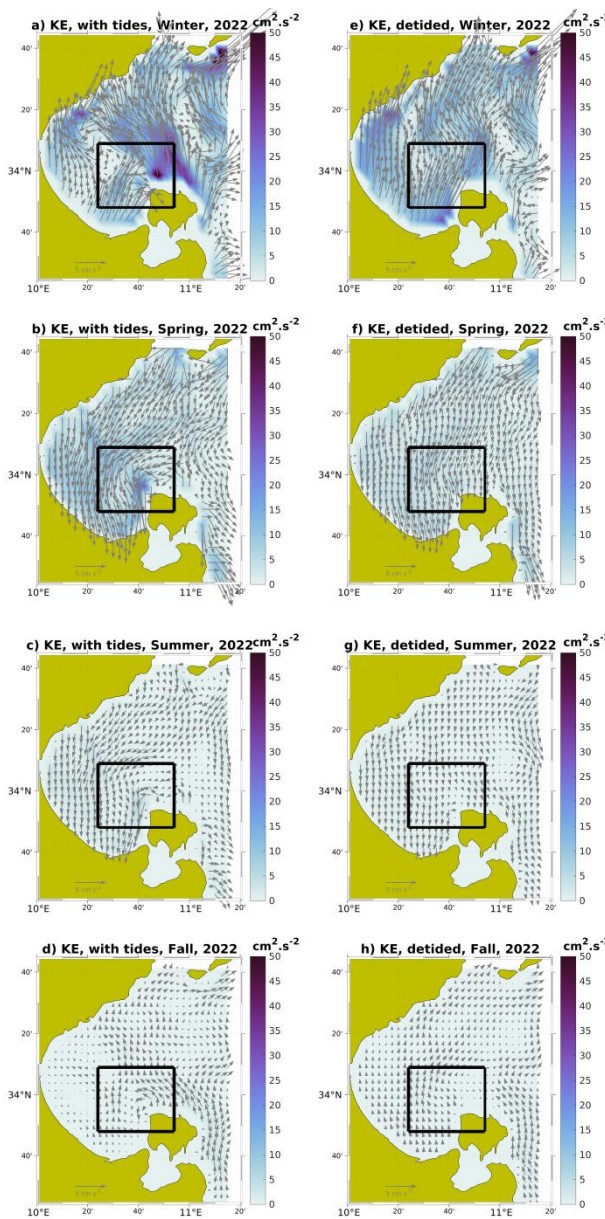

**Figure 5:** Seasonal mean geostrophic currents computed from MEDSEA_ANALYSISFORECAST_PHY_006_013 product SSH fields; both native with tidal forcing included (a-d) and detided (e-h) superimposed to seasonal Mean Kinetic Energy (MKE), in winter, spring, summer and fall for the year 2022. The average MKE values are shown in the inserts. The black rectangles show the regions where tides impact on the circulation in north-eastern part of the Djerba Island.

We quantified the impact of tides on the dynamics of the GGSD by computing the normalized Okubo-Weiss parameter Q* (eq. 6) from the geostrophic currents derived from the model SSH fields in 2022. The results are displayed in Figure 6 a-d



for the full signal and Figure 6 e-h for the tidal residual. Analyzing the geostrophic circulation derived from full SSH fields in the GGSD, the study area can be classified as a hyperbolic region throughout all the seasons, with a value of Q* clearly close to 1 for most of the time (>90%), due to strong deformation gradients. On the contrary, the diminished effect of

rotation on GGSD topology can be clearly observed for a small portion of the study area (<8%), where values of Q* are close to -1.

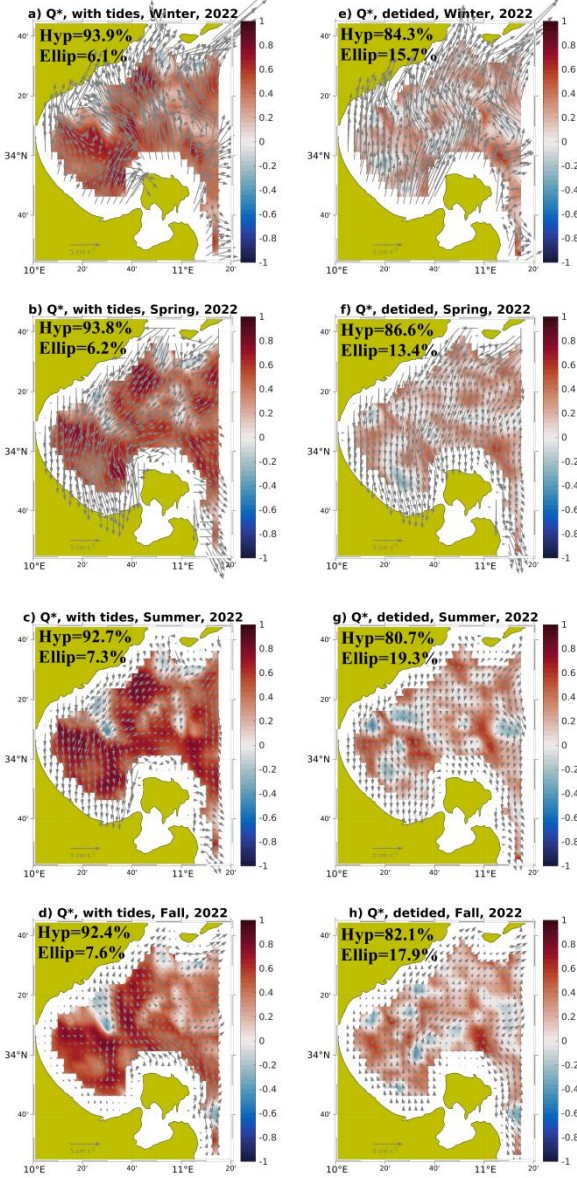

**Figure 6:** Seasonal mean normalized Okubo-Weiss parameter Q* with superimposed mean geostrophic currents computed from MEDSEA_ANALYSISFORECAST_PHY_006_013 product SSH fields, both native with tidal forcing included (a-d) and detided (e-h),

for the year 2022. The inserts show the percentage of the domain where Q*<0 (elliptic grid cells) and Q*>0 (hyperbolic grid cells).





The highest elliptic grid cell percentages were detected in summer (7.3%) and fall (7.6%) where the flow becomes more meandering than in winter and spring. In absence of tidal forcing a decrease of ~10% of the hyperbolic areas (Figure 6 e-h) can be observed. In contrast, an increase of ~10% of elliptic regions can be noticed, with the flow becoming meandering. In general, the GGSD is dominated by hyperbolic regions, and the presence of tides clearly enhances this tendency by approximately 10%. A possible explanation of the dominating hyperbolic structures even in the case of absence of tides might be related to the impact of wind on geostrophic circulation. Similar results were found in the Black Sea (Bouzaiene et al., 2021) where the hyperbolic regions are strongly dominant in winter due to larger wind stress, while in summer the elliptic areas are more pronounced because of the meandering currents.

**3.2.2    Impact of tides on strain and turbulence**

In order to confirm that tidal forcing is amplifying the deformation rate S* that could increase the recorded hyperbolic grid cells found in Figure 6, we computed the normalized deformation rate both from full SSH(case 1) and detided SSH (case 2) fields and the corresponding time series are shown in Figure 7. In the presence of tides (blue dots) the S* is larger than in the absence of tides (red dots) throughout the entire considered period. The difference between S* in the two cases has an average of approximately 0.1 (yellow dots) meaning that tides produce stretching/deformation rates in the GGSD and potentially enhance the presence of hyperbolic regions.

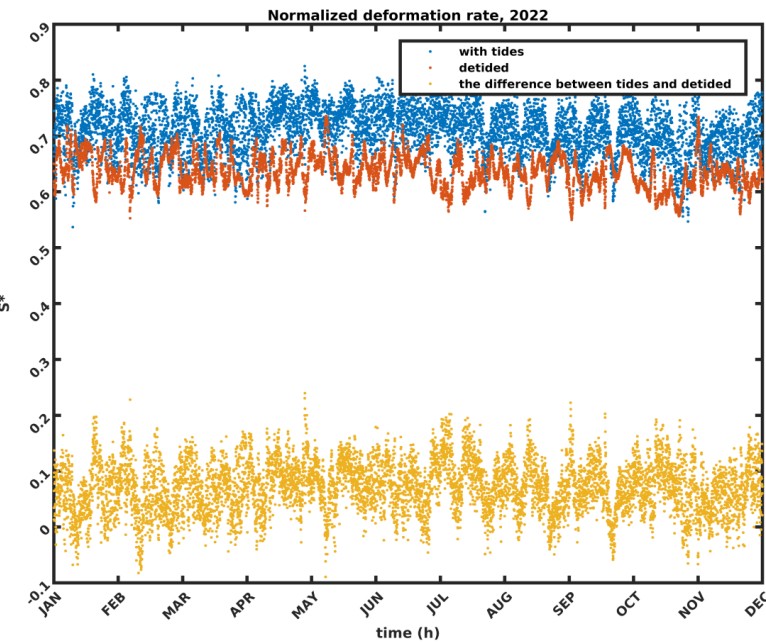

**Figure 7:** Time series of the deformation rate (S*) computed from MEDSEA_ANALYSISFORECAST_PHY_006_013 product in 2022: full SSH fields (blue curve), detided SSH fields (red curve) and their difference (yellow curve).





To investigate tides influence on GGSD, turbulence the Probability Density Function (PDF) of the normalized vorticity ($\zeta^*$) has been computed on a seasonal basis in the two different cases mentioned above and the results are shown in Figure 8. In the case of a 2D theoretical isotropic and homogenous turbulent flow, PDF shows a Gaussian shape without intermittency (absence of tails).

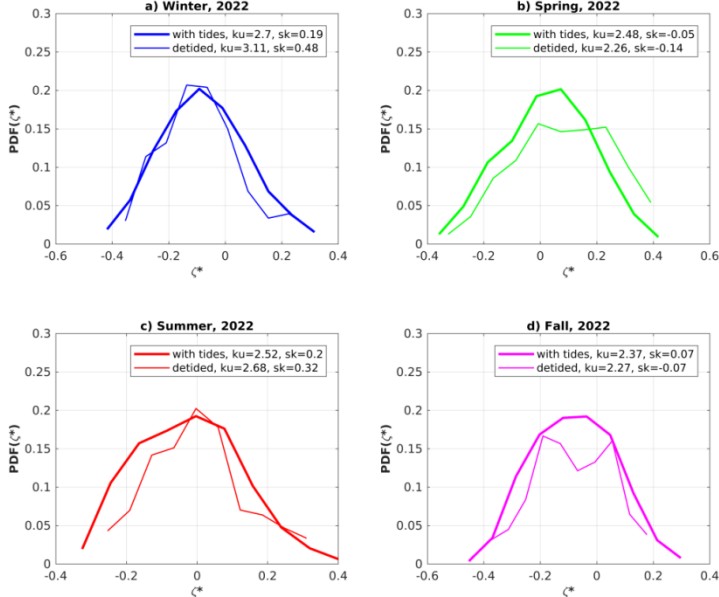

**Figure 8:** Seasonal Probability Density Function (PDF) of normalized vorticity ($\zeta^*$) computed from MEDSEA_ANALYSISFORECAST_PHY_006_013 product in 2022: full SSH fields (bold curves) and detided SSH fields (thin curves) in (a) winter, (b) spring, (c) summer and (d) fall.

In our study the PDFs exhibit nearly Gaussian shapes. These appear to be regular shapes for all the seasons in case of flow
affected by tides, with observable intermittency (long tails, ~-0.4 to 0.4) associated to the presence of coherent structures. The first case is well known as 2D quasi-geostrophic turbulence. Different Kurtosis and Skewness values depending on the season have been found. Low skewness values were detected in spring (sk=-0.05) and in fall (sk=0.07). Larger values were found in winter (sk=0.19) and summer (sk=0.2), with corresponding Kurtosis values of 2.48, 2.37, 2.7 and 2.52, respectively. The difference in Skewness and Kurtosis is essentially due to: (1) non-homogenous and anisotropic turbulent flow, (2) the
influence of coherent vortices, (3) the presence of stretching/deformation. The anisotropic flow for asymmetric PDFs may be due to the asymmetry in flood and ebb tidal currents (Song, et al., 2007). For the 2nd case (i.e., the non-tidal forcing GGSD dynamic) the shapes of all the normalized vorticity PDFs are irregular.

One of the possible applications of the results of our work is the study of the dispersion of nutrients connected to phytoplankton bloom events occurred in the GG and captured by NASA satellite images in March 2013 and 2017 as shown





in Figure 9. The concept behind this application is to enhance the knowledge on how physics drives biogeochemistry: by comparing the phytoplankton blooms to the $\lambda_t$ integrated over 7 days for high resolution of initial particle grid positions of 1/128°, this method may effectively enhance our comprehension of the link between physical processes and biogeochemistry. The qualitative correlation between algal bloom and $\lambda_t$ shows how this latter can be used as a proxy for the distribution of the biomass and nutrients within the gulf. The positive divergence detected in the central Gulf of Gabès (Fig.

4b) explains the tendency to upwelling in this area (Poulain, 1993). The surface chloro.phyll concentration from CMS multi-satellite observations at 1 km resolution is displayed in the inserts of Figure 9 along with the phytoplankton blooms captured from NASA (March 12, 2013, March 23, 2017). A link between physics (FTLE) and biogeochemistry (Chl-a bloom) is noticeable, with the chl-a being dispersed on the edges of the GG coherent structures.

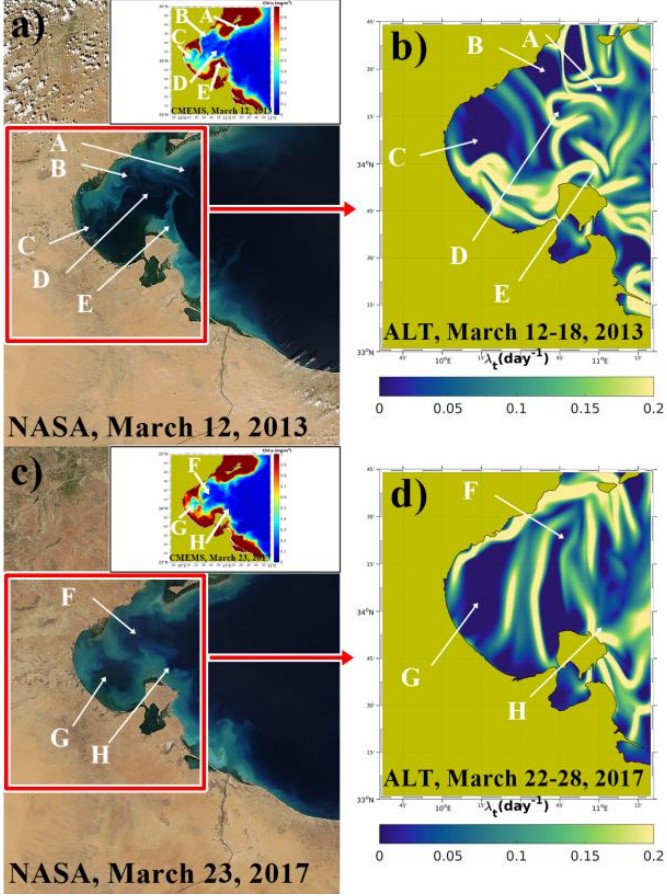

**Figure 9:** (a) Phytoplankton bloom off the coast of Tunisia-NASA MODIS Image of the Day, March 12, 2013, March 23, 2017 (https://modis.gsfc.nasa.gov/). (b) Spatial distribution of FTLE $\lambda_t$ computed from altimetry data (SEALEVEL_EUR_PHY_L4_MY_008_068) for 12-18 March 2013. (c) Phytoplankton blooms in the GG from NASA on March 23, 2017 and (d) spatial distribution of FTLE for 22-28 march 2017. Labels as A, B, C, D, E, F, G, H denote the features detected by NASA. In the insets we show chlorophyll a concentration from the European Union-Copernicus Marine Service (2022) for the same period as

phytoplankton blooms were captured by NASA.



This area is therefore the natural location for a high concentration of nutrients, favored also by the presence of cyclones (Salgado-Hernanz et al, 2019). GG features appear as transport barriers for phytoplankton dispersion. The phytoplankton blooms are driven by fronts, filaments, and mesoscale structures labeled as A, B, C, D, E, F, G and H in Fig. 9. A similar scenario has been observed in the northeastern Atlantic Ocean, where phytoplankton spring blooms are modulated by the

mesoscale dynamics. It has been found that mesoscale stirring due to the geostrophic circulation generates mesoscale chlorophyll structures (Lehahn, et al., 2007) in that region. The study shows that $\lambda_t$ is a good indicator of phytoplankton blooms.

## 4    Summary and conclusions

In this work the main hydrodynamic features of the Gulf of Gabès were investigated by means of geostrophic velocities

derived from 30 years of satellite altimetry data and 1 year (2022) of hourly SSH fields produced by a high resolution oceanographic numerical system. Altimetry analysis of the GG is presented in terms of normalized vorticity ($\zeta^*$), normalized divergence ($\delta^*$), normalized Okubo-Weiss ($Q^*$), normalized deformation ($S^*$) and FTLE ($\lambda_t$). The mean spatial distribution of $\lambda_t$ (1993-2022) confirms the presence of well-known features in the study area, such as ATC, MG, SMG and LSBV. The signature of these features can be found in the intensity of $\lambda_t$ (Fig. 2b). The FTLE is a powerful diagnostic tool for ocean

turbulence and horizontal mixing/stirring which vary inversely with phytoplankton concentration (Hernandez-Garcia et al., 2010). This may explain the poverty in nutrients in regions relatively far from the GG where we detected high values of $\lambda_t$, and it is in agreement with the known high biological production in areas close to the coast, where we found low values of $\lambda_t$. The CGG subarea can be considered a zone rich in nutrients (since $\delta^*$ showed some positive values) where the flow tends to spread particles. The application of $\lambda_t$ method on altimetry data is shown to be a key tool to understand how GG dynamics

drive passive organic tracers or pollutants. This scenario is observed for some seasons, with the divergence positive curve indicating the divergence of the CGG flow and explaining the phytoplankton blooms previously observed by Feki-Sahnoun et al. (2018). Furthermore, in the coastal zones located in the northern Gulf of Gabès (NGG) the flow tends to be neutral ($\delta^* \sim 0$), except for some seasons when $\delta^*$ is larger than zero, maybe due to the presence of upwelling events close to Djerba Island. In contrast, in the southern region of the Gulf of Gabès (SGG), the flow is convergent, since it is characterized by a

negative value of $\delta^*$. In this latter case the vorticity is concomitantly negative due to the signature of an anticyclonic vortex. Geostrophic coastal currents and eddies are associated with the presence of hyperbolic regions ($Q^*>0$) in any season, with a crucial role played by stretching/deformation gradients (Figs 3 and 4). To the best of our knowledge, this paper for the first time discusses how tides affect the GGSD flow topology (i.e., the distribution of elliptic and hyperbolic regions). The flow topology has been related to anomalous absolute dispersion regimes in the Mediterranean Sea (Bouzaiene, et al., 2018) and

the Black Sea (Bouzaiene, et al., 2021). Tides amplify hyperbolic regions in the GGSD, with more than 90% of hyperbolic grid cells ($Q^*>0$) captured in any season, while elliptic regions almost disappear due to the outgrowth of hyperbolic ones. We also observed a significant change in geostrophic circulation pattern through the different seasons, as discussed in Fig. 5. These different patterns could be related toulf topography, current instability and horizontal pressure gradients influenced by other atmospheric components. Lower MKE values are recorded in summer and autumn, while higher values are detected





in winter/spring. Since marine species disperse in regions characterized by lower turbulence and less energy intensity, our findings are in good agreement with the results found in Solgado Hernanz et al. (2019), where enhanced chlorophyll distribution in the GG starts in June, peaks in September and terminates in February.

Another finding of this study is the way tides contribute to stretching/deformation (S*) to define the signature of the hyperbolic regions. The time series of S* derived from geostrophic currents computed from the full SSH fields shows larger
values with respect to the one computed from detided SSH fields. This confirms that tides are dynamically responsible for the amplification of deformation gradients. PDFs emphasize the non-homogeneity and anisotropy of the GGSD turbulent flow due to the presence of eddies and intense strain factor.

To conclude, altimetry data available for the period of 1993–2022 was analyzed in the larger Gulf domain by comparing the dynamics in three subareas and defining their common characteristics. The central part of the gulf is dominated by an
upwelling flow (thus characterized by divergence) where there is an important biological production rate. The other areas located in its northern and southern parts are dominated by anticyclonic structures. The three subareas can be classified as hyperbolic regions (Q*<0). The high spatiotemporal resolution of model outputs allowed to analyze also the GGSD over the year 2022. In good accordance with the altimetry analysis, the GGSD can be classified as hyperbolic area with a signal greater than 90%, whereas the signature of elliptic zones is lower than 10%. Atmospheric conditions, topography and tides
forcing are very important for the occurrence of hyperbolic regions driven by strain. As well as amplifying hyperbolic regions, tides also affect the isotropy and homogeneity of the theoretical turbulent flow. PDF is applied to quantify the impact of tides on 2D turbulence theory and the PDF asymmetric distributions reveal the non-homogeneity and anisotropy of the surface flow due to persistent stretched currents and eddies.

*Data availability.* This study has been conducted using E.U. Copernicus Marine Service Information. The products used are MEDSEA_ANALYSISFORECAST_PHY_006_013                                  (Clementi                                  et al.2023,https://doi.org/10.25423/CMCC/MEDSEA_ANALYSISFORECAST_PHY_006_013_EAS8), SEALEVEL_EUR_PHY_L4_MY_008_068        (European        Union-Copernicus        Marine        Service        2021, https://doi.org/10.48670/MOI-00141)                and                OCEANCOLOUR_MED_BGC_L4_MY_009_144
((https://doi.org/10.48670/moi-00300, Mediterranean Sea Ocean Satellite Observations, the Italian National Research Council (CNR – Rome, Italy)) which are made available through the Copernicus Marine Data Store (https://data.marine.copernicus.eu/products).

*Author contributions.* Formal analysis, conceptualization, methodology, investigation, writing – original draft preparation:
MB. Data curation: AG. Funding acquisition: SS. Writing – review and editing: MB, AG, DD, AFD, SS and CF. All the authors have read and agreed to the published version of the paper.

*Funding.* This research was funded by the Italian Ministry of University and Research as part of the NextData project.



*Competing interests*. The authors have no competing interests.


*Acknowledgments*. We thank G. Haller and K. Onu for providing the code for FTLE computationin this study. Special thanks to my father, I am grateful for the many years you were my guiding star, my best gratitude to you, I will never forget your kindness and support even after your death.

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
