# Peer review of "Geostrophic circulation and tidal effects in the Gulf of Gabès"

_EGUsphere, 2024_

## Author Comment (AC3)

[Figure]

**Figure 1. Seasonal variability of wind stress at 10 m above the Central Gulf of Gabes (CGG) in 2022 computed from ECMWF data (1/8º) superimposed to seasonal wind circulation, in (a) winter, (b) spring, (c) summer and (d) fall.**

---

## Author Response (AR1)

We thank Referee #1 for his/her comments on the first version of the manuscript. We have addressed all the comments to improve the paper. Our responses to questions are detailed as follows:

Review of manuscript: Geostrophic circulation and tidal effects in the Gulf of Gabès by Bouzaiene et al

This study is about the ocean dynamics and kinematic properties of the Gulf of Gabès, with a focus on the influence of tides on circulation patterns and transport processes. The authors present very relevant research questions, and on an understudied region of the Mediterranean Sea. An interesting framework to investigating the tidal signals is presented too. While the study presents a well-motivated analysis, the manuscript would benefit from a clearer aim with more concrete objectives. This seems to be clear at the beginning, but the results need refining and cohesion. Additionally, even if the proposed framework and diagnostics are very interesting, certain methodological aspects require further justification and refinement, particularly regarding the choice of datasets, diagnostics and/or region of study one the research question is clearer. For example, if the idea is to focus more on the underlying dynamics that affect the phytoplankton blooms, things should be organised differently than if the main focus is on the impact of the tidal signal. I recommend that this manuscript be considered for publication, provided the authors restructure the study and address the following major concerns:

**Major comments**

Q1: This study presents relevant research question for the oceanographic physics community, but also for the biogeochemical one due to the insights it can bring to understanding the nutrients and phytoplankton present in the study zone and ones with similar dynamics. However, the research question I feel is not clear. There seems to be a focus on impact of tidal signal on the geostrophic circulation, but then a focus on phytoplankton impact, and how FTLEs can show this. Maybe linking better the ideas an results and help also make the research clearer and better linked to results, their discussion and conclusions.

R1: The aim of the present study is to focus on both geostrophic circulation in the Gulf of Gabès bigger domain (GG) and on the impact of tidal signals on geostrophic patterns in the Central Gulf of Gabès (CGG). The CGG represents also a site of particular interest to investigate the influence of physical processes on the biogeochemical features (with a focus on algal blooms). The focus on phytoplankton impact is mainly intended as an example of the application of metrics, such as FTLE, that can be used to describe ocean turbulence and horizontal mixing/stirring, but which can also be applied to other areas of marine sciences, fostering a better understanding of the links between physical and biogeochemical processes.

Q2: Throughout the manuscript it is mentioned several times without tidal forcing, when, to my understanding, what is removed is the tidal signal, but the impact of having tidal forcing on the geostrophic field is still there. If the direct impact of tidal forcing was the focus, a simulation without and with tidal forcing would be necessary.

R2: In the manuscript we focused on the impact of the tidal signal and that's the removal of the tidal signal that we refer to when we mention SSH detided fields. We agree that referring to this as without tidal forcing is misleading, thus we corrected this issue on the present version of the manuscript.

Q3: If the focus is on the effect of tides, it is not clear to me why there is so much focus put on the analysis of the altimetry data. The limitations are mentioned in L65. Again, a rearrangement of some ideas and analyses maybe could help clarify the relevance of the analyses with altimetric data, for example as a geostrophic baseline of what can be understood in the region of interest with these observations. Moreover, the reasoning behind looking at the 30 years of data is not clear. Would be interesting also to see an equivalent analysis to that done with the model data. It is not clear why both datasets used in this study are not compared. Lastly, there is no mention either on the impacts of their different resolutions (1/8 altimetry, 1/24 model).

R3: In the first part of the present manuscript we used a 30-years altimetry dataset with low spatial resolution (1/8 deg) to have an overview of the main circulation features (and their seasonal variability) of the Gulf of Gabès. The mean kinetic energy and FTLEs computed from the altimetry data show in the gulf larger domain the presence of well-known gyres, eddies and currents (MG, SMG, LSBV, ATC, AIS and BATC) that could influence the main circulation patterns of the GG smaller domain. Furthermore, FTLE, Div and O-W spatial distributions show the presence of three different dynamical areas close to the GG coastal zones; North Gulf of Gabes (NGG), Central Gulf of Gabes (CGG) and Southern Gulf of Gabes (SGG). The CGG dynamic is very different with respect to the NGG and SGG by the presence of filaments while both NCC and SGG were identified anticyclonic patterns. The CGG can be classified as an area with larger filaments. Since high resolution in situ and satellite data observations are limited in the CGG, especially close to boundaries not allowing to study tidal signal impact on geostrophic circulation. We used model data with high spatial (1/24 deg) and temporal (1 h) resolutions to focus on tidal signal impact on geostrophic pattern. The comparison of the geostrophic circulation from daiyled altimetry product and model data with resolutions of 1/8° and 1/24°, respectively in the GG bigger domain in 2022 is shown in Figure 5. We chose the year 2022 because, at the time the dataset was processed, it was the only complete (Jan-Dec) year for the MEDSEA\_ANALYSISFORECAST\_PHY\_006\_013 product, the only CMEMS product for the Mediterranean Sea that included the tidal forcing in the hydrodynamic model used. The general circulation features are retrieved from both observations and model in 2022 as follows: SMG, LSBV, ATC1, ATC2, ALC, AIS and BATC (Figure 5 a vs b). The impact of high resolution model data can be observed in ATC1 intensity in CGG (Figure 5 b).

Q4: More refinement needed: The choice of the temporal period, and of the spatial domain chosen is not clear. The same domain for the altimetry and model data is not chosen, when there is seems to be data available for the same domain. Moreover it worries me that features so close to land are studied given the limitations of both datasets in coastal areas. I understand the difficulty of having data to validate these datasets, and specially so close to land, but there is no mention of it in the datasets description, not even in the discussion. There is no mention of general validation of the datasets, for the model only the QUID. Should be clarified if for example that is

part of the reason to include the altimetry data, as a kind of ground truth to the model data (at a geostrophic level).

R4: For the computation of the mean kinematic features in the Gulf of Gabès using altimetry data, we selected a long time period (30 years) to ensure the robustness and representativeness of the analysis. For the model data, we chose the year 2022, as at the time the dataset was processed, it was the only complete available (Jan-Dec) year MEDSEA\_ANALYSISFORECAST\_PHY\_006\_013 product. This is the only CMEMS product for the Mediterranean Sea that includes the tidal forcing in the hydrodynamic model. Following the analysis of the physical features of the Gulf of Gabès, we focused on its central portion (CGG), where the highest tidal ranges are observed (Abdennadher and Boukthir, 2006; Othnani et al, 2017), in order to closely examine the area of the study region where the most significant impact of the tidal signal could potentially be detected. We agree that a specific validation of both the model and altimetry datasets for the study area is lacking. Nevertheless, both datasets have been extensively validated within the framework of their respective Quality Information Documents (QUIDs). In particular, see SEALEVEL EUR PHY L4 MY 008 068 (https://documentation.marine.copernicus.eu/QUID/CMEMS-SL-QUID-008-032-068.pdf), which we referenced in the updated version of the manuscript (line, 94). MEDSEA ANALYSISFORECAST PHY 006 013 (https://documentation.marine.copernicus.eu/QUID/CMEMS-MED-QUID-006-013.pdf), refer to pages 8-9 of the QUID. For this product, please refer specifically to the metrics of Region 7, which includes our study area (see Figure 1 on page 7 of the QUID). It is important to note that Region 7 is the second lowest in terms of the mean number of available SLA satellite observations per week (after the Northern Adriatic Sea), and it also has the highest SLA RMSD the regions listed in **Table** of MEDSEA\_ANALYSISFORECAST\_PHY\_006\_013 9). **QUID** This (page limitation is emphasized more explicitly in the present version of the manuscript, lines: 462-464. In Fig. 5, we included a comparison between the geostrophic currents derived from the altimetry product and those from the model product for the overlapping period, in order to demonstrate that the main circulation features are well represented by the model.

Q5: Missed part mentioning that once tidal forcing included (and also the fact that it is a wave-couple model), in some cases you might not be in geostrophic balance anymore. Also, having a higher resolution model, might also imply that the geostrophic balance does not dominate all the time anymore.

R5: We agree that, in regions or time scales where geostrophic components become significant, given that the model includes tidal forcing and wave coupling, we might not be in geostrophic balance anymore. Whereas, in the Mediterranean Sea, given that the high resolution model includes waves and tides, it offers an accurate geostrophic circulation (Escudier et al, 2021). lines: 464-467.

Q6: General formatting: wrong numbering of sections and subsections e.g. introduction should be 1, L138, ...

**R6: Done,**

Q7: Summary and conclusions needs improvement, better structure this section to make it clearer to reader. A lot of interesting points, that a better structure can help to understand and follow the ideas.

R7: Done,

**Specific comments**

L19: Add references here

Done, line: 20.

L20: "region of relevant tides": Add references. Moreover, as read later, it has relevant tides within the Mediterranean, but not compared to other regions globally. Please clarify text.

Done, line: 21.

L24: Add reference

Done, line: 25.

L54: "Two anomalous..": This paragraph is not very clear to me, namely why do you refer to these 2 regimes as anomalous? With respect to what?

The two anomalous or abnormal absolute dispersion regimes are referred with respect to two other well-known absolute dispersion regimes. These regimes were not studied theoretically till the study presented in Elhmaidi et al, 1993. They designed these two regimes as anomalous or abnormal regimes with respect to the ballistic t2 and random-walk t1 regimes.

L59 "shared": sheared?

Yes, sheared. We corrected this typo in the revised manuscript (line 61).

L60: There is preprint on this topic in other region: Gomez-Navarro et al: <a href="https://doi.org/10.1002/essoar.10512397.5">https://doi.org/10.1002/essoar.10512397.5</a>

In Gomez-Navarro et al, (2024) the impact of tidal forcing on surface particle transport is explored, while how tidal perturbation influences the dispersion of elliptic and hyperbolic regions lacks, in our opinion, a certain degree of discussion (lines: 61-63).

L72: Not directly tides, but Barkan et al., (2017, 2021), mention impact of internal waves (signal increases significantly when tidal forcing present) on mesoscale eddies.

**Done, lines: 75-76.**

Other references not mentioned that could be relevant:

- Drillet et al (2019)
- Ruhs et al (2025) (similar dataset used, but for impact of waves, important to mention here too as wave-coupled)

**References and text are added to the revised version of the manuscript (Lines: 76-78).**

L73: "estimated as the balance of the Coriolis force and the horizontal pressure gradients": maybe not necessary to include this here?

**Ok, the sentence is removed (line: 80).**

L82: Altimetry data

- Temporal resolution of data not mentioned
- daily (see please line 92 of the revised version of the manuscript).
- "EUROPEAN SEAS GRIDDED L4 SEA SURFACE HEIGHTS AND DERIVED VARIABLES REPROCESSED (1993-ONGOING) [dataset].": no need for uppercase and [dataset]. Improve reference to data.
- Done, lines 90-92.
- "30-year period (1993–2022)": line above states that ongoing? Please clarify
- Done, lines: 92-93.
- "variable used is the absolute surface geostrophic velocity, while altimetry data were used to estimate the vorticity,": this is not clear. Absolute surface geostrophic velocity is also inferred from altimetry data. Do you mean you inferred vorticity and the other parameters from this velocity variable of from the ADT or SLA?

Yes, that means we inferred vorticity and the other parameters from this velocity variable deduced from the ADT. The sentence is modified in the revised version of the manuscript (lines: 94-96).

L90: Chlorophyll-a data

- Should be chlorophyll-a??
- Yes, chlorophyll-a, corrected (line 99 of the revised version of the manuscript).
- Temporal resolution of dataset is daily? Please clearly specify
- Temporal resolution is daily (line 100 of the revised version of the manuscript).
- Missing brackets at end

**• Yes, brachets is added (line 101 of the revised version of the manuscript).**

L94: In the introduction you mention the model has temporal resolution, but this detail not included here.

**Specified at line 103: Hourly.**

L96: "We have chosen year 2022 since at the time the dataset was processed it was the only complete year for the CMS system including tidal signal in the hydrodynamic model used.": Related to general comment 4, if for the model data you were limited to year 2022, and given that the model includes data assimilation, why are the fields not compared to the altimetry fields during 2022 instead of the average of 30 years? (See major comment 3)

We added a new Figure 5 to compare altimetry fields to model outputs in 2022. The average of the 30 years altimetry data is functional for obtaining an overview of the geostrophic patterns in the GG.

L99 "coupled hydrodynamic-wave model": importance of being coupled with a wave model is not mentioned. This can also be affecting the geostrophic field as shown by other studies (Morales-Marquez *et al 2023*, Ruhs *et al*, 2025). Even if the focus here is on tides, I was expecting a mention to this important factor at least in the discussion.

**We agree, please see lines: 461-462 of the revised version of the manuscript.**

L105: "from the MEDSEA\_ANALYSISFORECAST\_PHY\_006\_013 product SSH fields": for clarity refer to this as model data including in brackets the product reference if you want, so that in L106 it does not seem that there 3 datasets

**Done, line 113 of the revised version of the manuscript.**

L106: "normalized": with respect to ?? Later you specify that to f and cite plain et al 2023, but this should already be clear here.

**Yes, "normalized": with respect to f (see lines: 114-115).**

L111: "the geostrophic equations as follows (Vigo et al., 2018a; 2018b)": maybe other references are more relevant? if not include as e.g.

**We added another reference, Apel (1987), line 119 of the revised version of the manuscript.**

L112: "sea surface elevation": specify (model SSH)

**Ok, modified line 122 of the revised version of the manuscript.**

L121: "Where H denotes the sea level elevation": so this is SSH too, i.e., ŋ? If so, please homegenize.

Yes, SSH stays for n, now it is more homogenized, lines: 131-132 of the revised version of the manuscript.

L122: Further details on the implementation of the deciding on the model data would be appreciated. For example to clarify the impact (if any) of the choice of parameter(s) in the detided result.

We used a low-pass symmetric filter to remove the tidal energy at diurnal and higher frequencies from model SSH for 39 hours of data for each value calculated. The filter is applied for each day. The choice of these parameters allows obtaining accurate detided SSH datasets as have been shown in the Manual on Sea Level Measurement and Interpretation of the IOC (1985) (https://psmsl.org/train\_and\_info/training/manuals/ioc\_14i.pdf).

L126: "sub-mesoscale, mesoscale, filaments, eddies and fronts activity": concepts mixed, please clarify

**We rephrased it, see please line 136 of the revised version of the manuscript.**

L127: The mentioned normalised vorticity would not be equivalent to the Rossby number? There is no mention of it and no references with respect to the order of magnitudes implying a mesoscale or submesoscale driven circulation, e.g. Thomas et al, (2008). Moreover, in this article they mention that for mesoscale Ro <<1 and O(1) for the submesoscale.

Yes, the normalized vorticity is equivalent to the Rossby number and we modified the definition (lines: 134, 136-138 in the revised version of the manuscript).

L140: Q\* is supposed to be normalized by f too? Then in eq. (6) you use S\* and  $\zeta$ \*?

**Yes,**

L145: Space missing after comma

**Ok,**

L149: "S is normalized by f to identify the sheared and/or stretched regions:" why need to normalize to show these regions?

S is normalized by f in order to get a dimensionless number, which represents a particularly effective tool for identifying sheared and/or stretched regions.

The Finite Time Lyapunov Exponents

■ L154: "In previous investigations within the Mediterranean region, the emphasis was on the Finite Scale Lyapunov Exponent (FSLE) rather than the FTLE." Aren't both FTLE and FSLE supposed to be equivalent? They should render the same (or very very similar) transport barriers. The only difference should be how the Lyapunov Exponent is calculated (defining time or space). The later mentioned gap could then be focused on calculating it in coastal areas, not the use of FTLE itself. As mentioned

in general comments, it is important to consider that the implementation of this in coastal areas, namely from altimetry data, has been limited by the error of the data in very coastal areas.

- Yes, FTLE and FSLE are very similar since both are detecting LCS. However, the difference between them is in the calculation methods. FSLE is deduced from the exponential growth of distances between Lagrangian particle pairs initially separated by a predefined distance while FTLE is computed from the separation rate of initially neighboring particles for a finite time. The novelty of this paper is to focus on FTLE within the Gulf of Gabès coastal areas. The implementation of FTLE in coastal areas, namely from altimetry data, has been limited by the error of the data in very coastal areas (line 170-171).
- Integration time of 6 days? FTLE fields are then obtained daily? And averaged for 30 days and 7 days? Please clarify.
- FTLE fields are obtained daily and then averaged seasonally over a 30 year period (lines: 183-184 of the revised version).
- L166: missing tr "indicates the.."
- Done, line 177 of the revised version.
- L169: Missing clearer explanation that FTLE can be implemented forward and/or backward in time and the implications for phytoplankton as one shows attracting and the other repelling structures.
- Done, lines: 179-180 of the revised version.
- L172: "30 years to detect mean features": why 30 years? Are so many years necessary?
- The Mediterranean features are strongly driven by the instability of intense coastal currents, which have frequently changed their location and lifespan over the past decades (Bouzaiene et al, 2020; Poulain et al, 2012). In order to investigate the kinematic properties of mesoscale features, we used 30 years of altimetry data in the present paper, focusing on the main circulation features in the GG. This 30 year dataset allows for the detection of mean patterns across three decades, providing a basis to discuss the well-known mean features during the observational data availability period (lines: 201-206 of the revised manuscript.
- L175: "16 times larger": is it really necessary? I understand it is beneficial to go below the grid resolution, but I was expecting around 4 times more.
- 16 times larger to show a high resolution FTLE image.

L180: A 2D bathymetry figure could be used to compare the figures shown in the results where impact of bathymetry mentioned.

Done, see please Fig 1 of the revised version.

L190: Fig. 2 (top), and other figures showing currents should be bigger or refined to see better the circulation patterns mentioned in the discussion of the results.

**Done,**

L200: GGSD area could be shown in fig. 2, but difference in domain shown for altimetry and model data is confusing. Limitation mentioned here clear, but then why not same domain used for model data as for the altimetry results shown in fig. 2?

**We modified the GGSD to be CGG for the coherence of the results.**

L203: Are these averages also removed from the altimetry data?

**No,**

L205: Therefore, only model data really used to assess effect of tides? Makes it confusing then as to why altimetry data then used in this study.

**See please R3.**

L220: this could go in the methods section were the FTLEs are described.

Done, see lines: 190-192.

L265: Some of these details could go in the introduction.

Done, see lines: 26-27.

L271: Can you give more details on this agreement?

Ok, our results are in good agreement with previous studies in the Mediterranean Sea (Vigo et al., 2018a) where in winter/fall the mean flow tends to inflow from the south to the north, while in spring and summer its circulation is mainly cyclonic bordering the coastline.

L285: Maybe missing something but not very clear for me in figure 5. Please give more details.

**We added in the new figs 6 and 7 red lines to show the presence of the cyclonic currents that are influenced by tides.**

L290: Some of the circulation patterns (for example cyclonic circulation in the mentioned cases) not very clear, maybe improvement of figure can help. Also colorer in last 2 rows might need to be adjusted?

**Done, please see the new figures, 6-9.**

L308: "absence of tidal forcing", would actually be without tidal signal not tidal forcing as detided. See major comment 2.

**Yes, without tidal signal, see line 348.**

L314: Maybe you can support this theory with wind data? Maybe the atmospheric forcing data used for the model?

The dominance of hyperbolic regions in the GG is deduced from Q\* which was derived from the surface current velocities. Looking at wind data used for the model does not allow us to focus on the direct impact of winds on the presence of the hyperbolic regions. We referred to Bouzaiene et al, (2021) where they related the presence of hyperbolic structures to wind stress in Black Sea. The wind is the most responsible factor driving surface circulation while tidal perturbations are very limited in that region.

L325: How is the PDF obtained? What is the sensitivity of the skewness and kurtosis values to this?

The PDF is obtained by computing the histogram of normalized vorticity as a function of season. The sensitivity of the skewness and kurtosis values in case of the presence of tides and detided one is to identify how tides impact turbulence being the anisotropy of the GG turbulent flow. We also showed as a function of season the skewness and kurtosis values are very different, which means that the GG dynamics is strongly influenced by seasonal variability of atmospheric forcing.

L330: Maybe adding a grid on the plot and having all x-axis alike can help make results clearer

**Done (see figure 11).**

L336: "The first case": ??

**The first case means SSH fields including tides, line: 376.**

L343: Application mentioned is very interesting, but the connection with previous results not so clear. Also the connection between FTLE and Chl-a plots a bit hard to understand. This last part needs improvement.

Done, lines: 383-391, line 416-422 and the new Figure 13 of the revised manuscript.

L515: Here year of publication put at end, check that formatting of references is consistent.

Done,

We thank Referee #2 for his/her careful reading of the first version of the manuscript and for his constructive remarks. Following his/her constructive comments, we tried to make the manuscript clearer.

The authors study surface circulation in Gulf of Gabes near Libyan coast in the Mediterranaen Sea. I was not familiar with this region even though I worked on several other regions in the Med domain. The study is conducted mainly using altimeter data. The primary original aspect of the study is that the effect of tides on FSLEs are studied. I was also not aware that tides were of any importance in the Med, but it seems this region has some of the largest tidal effects. The main conclusion of the study is that tidal effects increase the importance of hyperbolic regions, hence chaotic advection. I guess that it makes sense that moving around the hyperbolic region by tidal influences would do that.

I do not think that this is a major discovery, but the study region is probably undernalyzed and the study is well conducted. So for these reasons, I do not have a major objection to publication of this paper. It is good to have things documented in this way.

---

## Referee Report (RR1)

**Review of manuscript egusphere-2024-3730 entitled "Geostrophic circulation and tidal effects in the Gulf of Gabès"**

**Main comment:**

Within the manuscript the authors use a 30-years time series of altimetry data as well as a numerical model (all freely available from CMEMS, Copernicus Marine) to investigate the dynamics in the gulf of Gabès. This region is of particular interest since it is the area of exchange between Western and Eastern Mediterranean Sea water masses. The authors perform a climatological study of the geostrophic circulation and investigate the effect of tides leading to the generation of a cyclonic current. The effect of persistent Lagrangian structures (FTLE) on the phytoplankton bloom occurrence is also discussed.

The paper is detailed, well-written and well structured and I think provide a quite complete overview of the dynamics, as seen by altimetry (or limited only to geostrophic balance) of the area. In its current form the paper is very interesting but I think would benefit from few more information/analysis before it can be published. Therefore I would recommend to publish the manuscript after some major revision. Please find in the following my detailed comments.

**Major comments:**

- 1) Even though I am sure that this kind of climatological review is necessary for a good understanding of the studied area, it seems to me that the text lacks from any explanations about what this kind of analysis brings in terms of new knowledges. It stated several times the results agree with previous work but never what we are the additional information. For example in the Introduction and Conclusion, the authors may emphasize more on the novelty of their approach compared to previous studies. I really think this could boost the readers' interest.
- 2) One point that is not clearly stated in the entire text, although written on line 421, is that FTLEs are dynamical diagnostics allowing to identify frontal/stretching areas it cannot be used as a diagnostic of biogeochemical processes. They can explain the relative 2D horizontal dispersion/distribution of some biological quantities and thus provide some insights on potential vertical processes that may engender phytoplankton blooms (Lévy et al.,). I would like to draw the authors attention on the fact that throughout the text a confusion can arise especially in section 3.2.2 (see detailed comments). Also, the title of section 3.2.2 is a bit confusing tome. I would not talk about turbulence here for several reasons:
  - FTLE are not a diagnostic of turbulence, especially when computed with low-resolution altimetry-derived (geostrophic) surface currents
  - In the present study, the authors got interested in features detected by persistent FTLEs (a mean over a long time period) which means that the features discussed here occur at

temporal scales (years) that are way larger than turbulence (days) or even fine-scales (weeks-months).

I would thus recommend to modify the section title for something like "impact of tides on strain and effect on biogeochemical distribution" or even the authors may consider splitting tides and FTLEs into two different subsections.

3) The authors provide a quite complete overview of the dynamics in the Gulf of Gabès but never discuss the evolution of SSH (surface currents velocity or direction...) as monitored by the altimetry time series between 1993 to 2022. This could provide insights on the evolution of the regional dynamics (any trends?) in the context of the climate change. In the discussion section these trends (if any?) could be discussed for future years evolution and potential impact on biology.

**Detailed minor comments:**

- L 15: "biogeochemical processes": I would rather use "biogeochemical dispersion".
- L 24: "richest" for the Mediterranean Sea yes but it is relative for other "rich" places in the world ocean. Maybe the authors can cite some references here.
- L 35: "One of them ... southward ()." I could not understand this sentence, please rephrase.
- L 44: "spatial-temporal" change for "spatio-temporal"
- L 53: "exert" not sure if it is correct in English, "act"?
- L 163: You can also cite other types of applications such as: d'Ovidio et al. (2010), Rousselet et al. (2025).
- L169-171: I totally agree with these statements, however I don't see how in this study these gaps are leveraged? Please maybe add a comment in the text.
- L 180: forward in time.
- L178-192: I do not understand for how long are the particle trajectories advected to computed FTLE?
- Figure 1: I think only two panels would be sufficient (either 2D or 3D bathymetry).
- L 207: Even though I agree with the theory, some subareas are very coastal and we know that altimetry is not really reliable there, so maybe the authors can justify the use of altimetry data in such coastal zones.

L 234-235: This is related to the major comment 2). Mean FTLE averaged over 30-year altimetry cannot be used to investigate chaotic turbulence since it is detecting large scale persistent (permanent) features. However I agree that such diagnostic is comparable to a mean concentration of Chl-a, I am just concerned by the sentence and reference to "chaotic turbulence".

L 259: "several cyclonic eddies". Again here can we rather talk about "permanent/recurrent eddies" or even "gyres"?

Figure 4: In the caption please specify that the quantities are mean over each boxes.

L 286-287: "the model results" at the surface. No comparison are performed on the vertical. Also is the model assimilating any observations? Because if the model is assimilating satellite data then the agreement between the model and observations is obvious and I think this part should be removed.

L 389-390: I don't understand how the comparison between Chl-a and FTLE can "provide insights into the time lag"?

L 391: "biogeochemical processes". I would change processes for "dispersion" since the biogeochemical processes are never really discussed (which one ? How?...)

L 414-415: I am not sure about this statement because many FTLE occurrences are not linked with any phytoplankton bloom (or more specifically high concentration of Chl-a). The authors should clarify or explain.

L 437-439: Here I would lin this to a dynamical process: "FTLE" act as barriers to offshore transport.

L 443-444: This statement is redundant, please remove or move to methods.

**References:**

d'Ovidio, F., De Monte, S., Alvain, S., Dandonneau, Y. and Lévy, M., 2010. Fluid dynamical niches of phytoplankton types. *Proceedings of the National Academy of Sciences*, *107*(43), pp.18366-18370.

Louise Rousselet, Francesco d'Ovidio, Lloyd Izard, Alice Della Penna, Anne Petrenko, et al.. A Software Package for an Adaptive Satellite-based Sampling for Oceanographic cruises (SPASSOv2.0): tracking fine scale features for physical and biogeochemical studies. 2024. <a href="https://doi.org/10.2016/na.2016.2016">https://doi.org/10.2016/na.2016.2016</a>

---

## Referee Report (RR2)

**Review of manuscript egusphere-2024-3730 ATC2 entitled "Geostrophic circulation and tidal effects in the Gulf of Gabès"**

**Main comment:**

The authors responded clearly to every of my comments. In my own opinion the paper is ready to be published as all analysis are coherent and consistently discussed. Therefore I suggest to accept the paper as is.

Please just note the new reference:

Rousselet, L., d'Ovidio, F., Izard, L., Della Penna, A., Petrenko, A., Barrillon, S., ... & Doglioli, A. (2025). A Software Package for an Adaptive Satellite-based Sampling for Oceanographic cruises (SPASSOv2. 0): tracking fine scale features for physical and biogeochemical studies. *Journal of Atmospheric and Oceanic Technology*. (Available on google scholar)

---

## Author Response (AR2)

We thank Referee #3 for her helpful and important comments on the revised version of the manuscript. We have addressed all the comments to improve the paper. Our responses to questions are detailed as follows:

Review of manuscript egusphere-2024-3730 entitled "Geostrophic circulation and tidal effects in the Gulf of Gabès"

**Main comment:**

Within the manuscript the authors use a 30-years time series of altimetry data as well as a numerical model (all freely available from CMEMS, Copernicus Marine) to investigate the dynamics in the gulf of Gabès. This region is of particular interest since it is the area of exchange between Western and Eastern Mediterranean Sea water masses. The authors perform a climatological study of the geostrophic circulation and investigate the effect of tides leading to the generation of a cyclonic current. The effect of persistent Lagrangian structures (FTLE) on the phytoplankton bloom occurrence is also discussed.

The paper is detailed, well-written and well structured and I think provide a quite complete overview of the dynamics, as seen by altimetry (or limited only to geostrophic balance) of the area. In its current form the paper is very interesting but I think would benefit from few more information/analysis before it can be published. Therefore I would recommend to publish the manuscript after some major revision. Please find in the following my detailed comments.

**Major comments:**

- **Q1**) Even though I am sure that this kind of climatological review is necessary for a good understanding of the studied area, it seems to me that the text lacks from any explanations about what this kind of analysis brings in terms of new knowledges. It stated several times the results agree with previous work but never what we are the additional information. For example in the Introduction and Conclusion, the authors may emphasize more on the novelty of their approach compared to previous studies. I really think this could boost the readers' interest.
- R1) Done, see please lines: 53-58; Some efforts have been made to focus on the dynamics of offshore waters in the central Mediterranean Sea from satellite-derived products, i.e. the dynamics in Sicily Channel show multi-scale spatial and temporal variability (Menna et al, 2019). Nevertheless, a long term analysis for understudied regions like the coastal GG area can benefit an overview of: persistent Lagrangian structures, attracting and repelling coastal zones, trends and upwelling flow.

The geostrophic circulation from altimetry data for the three decades within the GG is characterized by strong seasonal and spatial variability where the dynamics varies differently in the three subareas (lines: 609-611).

**Q2**) One point that is not clearly stated in the entire text, although written on line 421, is that FTLEs are dynamical diagnostics allowing to identify frontal/stretching areas it cannot be used as a diagnostic of biogeochemical processes. They can explain the relative 2D horizontal dispersion/distribution of some biological quantities and thus provide some insights on potential vertical processes that may engender phytoplankton blooms (Lévy et al.,). I would like to draw the authors attention on the fact that throughout the text a confusion can arise especially in section 3.2.2 (see detailed comments). Also, the title of section 3.2.2 is a bit confusing tome. I would not talk about turbulence here for several reasons:

- FTLE are not a diagnostic of turbulence, especially when computed with low-resolution altimetry-derived (geostrophic) surface currents
- In the present study, the authors got interested in features detected by persistent FTLEs (a mean over a long time period) which means that the features discussed here occur at 1 temporal scales (years) that are way larger than turbulence (days) or even fine-scales (weeksmonths).

I would thus recommend to modify the section title for something like "impact of tides on strain and effect on biogeochemical distribution" or even the authors may consider splitting tides and FTLEs into two different subsections.

R2) Done, see please lines: 532-534.

We totally agree with the proposition and we modified the section title accordingly (line 454).

**Q3**) The authors provide a quite complete overview of the dynamics in the Gulf of Gabès but never discuss the evolution of SSH (surface currents velocity or direction...) as monitored by the altimetry time series between 1993 to 2022. This could provide insights on the evolution of the regional dynamics (any trends?) in the context of the climate change. In the discussion section these trends (if any?) could be discussed for future years evolution and potential impact on biology.

R3) We computed the daily mean speed and kinetic energy (KE) time series over 30 years (1993-2022) using altimetry data, as shown in Figure 4. The quantities are averaged over the larger GG box, indicated by the red rectangle in Figure 2. Higher speed and KE values are mostly observed in winter and fall, while lower values occur in spring and summer. This variability is likely strongly related to atmospheric forcing. In order to evaluate the evolution of regional dynamics over the decades, we computed the means of the two quantities separately for the three following periods: 1993-2002, 2003-2012 and 2013-2022. The mean speed increased over the decades, from 7.35 cm/s in 1993–2002, to 7.6 cm/s in 2003–2012, and 8.01 cm/s in 2013–2022. Similarly to the averaged speed, the Mean Kinetic Energy also increased by approximately 7 cm²/s² from the beginning to the end of the considered period. See please lines: 324-334.

This study investigates sea surface height trends over the GG from 1993 to 2022, where the surface layer shows a speed trend of 0.033 cm/s and a KE trend of 0.34 cm²/s² as shown in Figure 3. The evolution of regional dynamics, and the consequent potential impact on biogeochemical aspects, is certainly a highly interesting topic, worthy of further investigation in future studies (lines: 555-559).

**Detailed minor comments:**

L 15: "biogeochemical processes": I would rather use "biogeochemical dispersion".

Done, Line 19.

L 24: "richest" for the Mediterranean Sea yes but it is relative for other "rich" places in the world ocean. Maybe the authors can cite some references here.

Done, see please line 31.

L 35: "One of them ... southward ()." I could not understand this sentence, please rephrase.

Done, see please line 45.

L 44: "spatial-temporal" change for "spatio-temporal"

Ok, line 60.

L 53: "exert" not sure if it is correct in English, "act"?

Ok, line 72.

L 163: You can also cite other types of applications such as: d'Ovidio et al. (2010), Rousselet et al. (2025).

Done, line 204.

L169-171: I totally agree with these statements, however I don't see how in this study these gaps are leveraged? Please maybe add a comment in the text.

This study seeks to address this gap by computing  $\lambda t$  specifically for LCS analysis in these areas. The implementation of FTLE using particle trajectories with increasing resolution (Onu, et al, 2015) in the GG could bring new insight into how coastal features impact biology. The use of FTLE in coastal areas is reliable to detect LCS (Peng et al, 2024). See please lines: 215-217.

L 180: forward in time.

Done, line 227.

L178-192: I do not understand for how long are the particle trajectories advected to computed FTLE ?

The particle trajectories are daily advected and then averaged seasonally over a 30-year period (lines: 231-232).

Figure 1: I think only two panels would be sufficient (either 2D or 3D bathymetry).

Done,

L 207: Even though I agree with the theory, some subareas are very coastal and we know that altimetry is not really reliable there, so maybe the authors can justify the use of altimetry data in such coastal zones.

Yes, the use of the altimetry data in very coastal areas can be limited by the lower spatial resolution of the data. Due to the lack of high resolution long term datasets availability in the GG we use altimetry data in its coastal areas. Whereas, altimetry analysis could help to overview long term kinematic properties in the coastal regions (Rinivasan and Tsontos, 2023) (lines: 281-282).

L 234-235: This is related to the major comment 2). Mean FTLE averaged over 30-year altimetry cannot be used to investigate chaotic turbulence since it is detecting large scale persistent (permanent) features. However I agree that such diagnostic is comparable to a mean concentration of Chl-a, I am just concerned by the sentence and reference to "chaotic turbulence".

Our intention was to highlight the spatial patterns of stirring as inferred from FTLE, so we agree that the term "chaotic turbulence" for a 30-year mean FTLE analysis is not proper, thus we modified "chaotic turbulence" to "GG dynamics" (see please line: 315).

L 259: "several cyclonic eddies". Again here can we rather talk about "permanent/recurrent eddies" or even "gyres"?

Done, see please line 355.

Figure 4: In the caption please specify that the quantities are mean over each boxes.

Done, please note that Figure 4 has become Figure 5 in the revised version of the manuscript.

L 286-287: "the model results" at the surface. No comparison are performed on the vertical. Also is the model assimilating any observations? Because if the model is assimilating satellite data then the agreement between the model and observations is obvious and I think this part should be removed.

Yes, the model is assimilating satellite data. We removed this part and the old Figure 5.

L 389-390: I don't understand how the comparison between Chl-a and FTLE can "provide insights into the time lag"?

The sentence is removed.

L 391: "biogeochemical processes". I would change processes for "dispersion" since the biogeochemical processes are never really discussed (which one ? How?...)

Done, we modified the sentences, see please lines: 494-496.

L 414-415: I am not sure about this statement because many FTLE occurrences are not linked with any phytoplankton bloom (or more specifically high concentration of Chl-a). The authors should clarify or explain.

Yes, we agree that some FTLE/FSLE occurrences are not usually linked with phytoplankton bloom. But in some cases of geostrophic currents FTLE/FSLE can show fronts producing Chl-a filaments controlling phytoplankton bloom. See please for more details (Lehahn et al, 2007; Guinder et al, 2025). See please lines: 522-524.

L 437-439: Here I would link this to a dynamical process: "FTLE" act as barriers to offshore transport.

Done, see please line 552.

L 443-444: This statement is redundant, please remove or move to methods.

Ok, the statement is removed.

References: d'Ovidio, F., De Monte, S., Alvain, S., Dandonneau, Y. and Lévy, M., 2010. Fluid dynamical niches of phytoplankton types. Proceedings of the National Academy of Sciences, 107(43), pp.18366-18370.

Louise Rousselet, Francesco d'Ovidio, Lloyd Izard, Alice Della Penna, Anne Petrenko, et al.. A Software Package for an Adaptive Satellite-based Sampling for Oceanographic cruises (SPASSOv2.0): tracking fine scale features for physical and biogeochemical studies. 2024. (hal-04705438)

---

## Author Response (AR3)

Dear Editor,

We are writing to express our sincere appreciation for considering our paper. Based on your suggestions we have made technical corrections to the paper based on your comments and the referee comments.

In response to your comments:

- 1) We updated the reference (lines 795-798)
- 2) We checked that the references are in alphabetical order.

We would like to extend our gratitude to you and the reviewers for providing us with insightful and constructive comments. We are grateful for the opportunity you have given us to publish our research in Ocean Science. Your guidance has immensely helped us refine our study.

Yours sincerely,

Maher Bouzaiene